# Alzheimer-mutant γ-secretase complexes stall amyloid β-peptide production

**Parnian Arafi[1], Sujan Devkota[1], Emily Williams[2], Masato Maesako[2], Michael S Wolfe[1]***

[1]Department of Medicinal Chemistry, University of Kansas, Lawrence, United States; [2]Alzheimer Research Unit, MassGeneral Institute for Neurodegenerative Disease, Massachusetts General Hospital, Harvard Medical School, Boston, United States

## eLife Assessment

This manuscript provides **fundamental** studies to gain insight into the mutations in the presenilin-1 (PSEN1) gene on proteolytic processing of the amyloid precursor protein (APP). The authors provide **compelling** evidence using mutations in PSEN to understand what drives alternative substrate turnover with **convincing** data and rigorous analysis. This deep mechanistic study provides a framework towards the development of small molecule inhibitors to treat AD.

*For correspondence:
mswolfe@ku.edu

Competing interest: The authors declare that no competing interests exist.

**Abstract** Missense mutations in the amyloid precursor protein (APP) and presenilin-1 (PSEN1) cause early-onset familial Alzheimer's disease (FAD) and alter proteolytic production of secreted 38-to-43-residue amyloid β-peptides (Aβ) by the PSEN1-containing γ-secretase complex, ostensibly supporting the amyloid hypothesis of pathogenesis. However, proteolysis of APP substrate by γ-secretase is processive, involving initial endoproteolysis to produce long Aβ peptides of 48 or 49 residues followed by carboxypeptidase trimming in mostly tripeptide increments. We recently reported evidence that FAD mutations in APP and PSEN1 cause deficiencies in early steps in processive proteolysis of APP substrate C99 and that this results from stalled γ-secretase enzyme-substrate and/or enzyme-intermediate complexes. These stalled complexes triggered synaptic degeneration in a *Caenorhabditis elegans* model of FAD independently of Aβ production. Here, we conducted full quantitative analysis of all proteolytic events on APP substrate by γ-secretase with six additional PSEN1 FAD mutations and found that all six are deficient in multiple processing steps. However, only one of these (F386S) was deficient in certain trimming steps but not in endoproteolysis. Fluorescence lifetime imaging microscopy in intact cells revealed that all six PSEN1 FAD mutations lead to stalled γ-secretase enzyme-substrate/intermediate complexes. The F386S mutation, however, does so only in Aβ-rich regions of the cells, not in C99-rich regions, consistent with the deficiencies of this mutant enzyme only in trimming of Aβ intermediates. These findings provide further evidence that FAD mutations lead to stalled and stabilized γ-secretase enzyme-substrate and/or enzyme-intermediate complexes and are consistent with the stalled process rather than the products of γ-secretase proteolysis as the pathogenic trigger.

## Introduction

Alzheimer's disease (AD) is a significant challenge to global public health as aging populations worldwide face its profound impact on individuals, families, and healthcare systems. The urgent need for accessible and effective disease-modifying therapies underscores the critical importance of unraveling the underlying mechanisms of AD (*Li et al., 2022*). For over three decades, the amyloid cascade hypothesis has been central in AD research, positing that aggregation of amyloid β-peptides (Aβ),

particularly the 42-residue variant (Aβ42), initiates a cascade of events leading to neurodegeneration and dementia (*Selkoe and Hardy, 2016*). This hypothesis emerged with the discovery of dominant missense mutations in the amyloid precursor protein (APP) associated with early-onset familial Alzheimer's disease (FAD) that alter Aβ production (*Hardy and Allsop, 1991*; *Selkoe, 1991*).

The subsequent discovery of FAD mutations in presenilin-1 (PSEN1) and presenilin-2 (PSEN2), catalytic components of γ-secretase complexes responsible for Aβ production from APP C-terminal fragment C99, further reinforced the amyloid hypothesis (*Selkoe and Hardy, 2016*). These mutations generally increase the ratio of aggregation-prone Aβ42 to more soluble Aβ40. However, the processing of Aβ is complex, involving multiple cleavages of the APP transmembrane domain (TMD) by the membrane-embedded γ-secretase complex, resulting in Aβ peptides through two distinct pathways: C99→Aβ49→Aβ46→Aβ43→Aβ40 and C99→Aβ48→Aβ45→Aβ42→Aβ38 (*Figure 1A*; *Takami et al., 2009*). Despite these advances, uncertainties persist regarding the assembly states of neurotoxic Aβ species and their roles in AD pathophysiology (*Benilova et al., 2012*). Moreover, clinical trials targeting Aβ or its aggregates have shown only modest efficacy, prompting a reevaluation of Aβ as the primary driver of the disease process (*Kepp et al., 2023*).

The clinical and pathological similarities between early-onset FAD and sporadic late-onset Alzheimer's disease suggest shared underlying disease mechanisms (*Bateman et al., 2011*; *Morris et al., 2022*). The monogenic nature of FAD, driven by APP or presenilin mutations, provides a clearer path to elucidating pathogenic mechanisms. FAD mutations in the APP TMD disrupt γ-secretase-mediated proteolysis: Our comprehensive analysis of effects on each proteolytic step for 14 such mutations demonstrated that the first and/or second carboxypeptidase trimming step was deficient in every case, elevating levels of Aβ peptides of 45 residues and longer (*Devkota et al., 2021*). Similarly, we found that six PSEN1 FAD-mutant γ-secretase complexes show reduced processive proteolytic function (*Devkota et al., 2024*). These PSEN1 mutations were all deficient in the initial endoproteolytic (ε) cleavage of C99 that generates Aβ48 or Aβ49 and the corresponding APP intracellular domain (AICD) coproducts, in addition to being deficient in one or more Aβ intermediate trimming steps. In *Caenorhabditis elegans* models of FAD, deficient processive proteolysis by FAD mutations was linked to stalled/stabilized γ-secretase enzyme-substrate (E-S) complexes that triggered age-dependent synaptic degeneration independently of Aβ production.

In the present study, we have expanded the comprehensive analysis of processive proteolysis of C99 by γ-secretase to include six additional PSEN1 mutations (S169L, S170F, G378E, F386S, A431E, A434T). Each proteolytic step was quantified by MS, and interactions between substrate C99 or intermediate Aβs and γ-secretase were measured by fluorescence lifetime imaging microscopy (FLIM). The results demonstrated that all six FAD PSEN1 mutations lead to reduction of multiple proteolytic steps while increasing the stability of E-S and/or enzyme-intermediate complexes, findings consistent with our 'stalled complex' hypothesis of AD pathogenesis. Three mutations displayed unusual profiles of effects on Aβ production that have novel implications for this hypothesis.

## Results

### FAD-mutant PSEN1 reduces processive proteolysis of C99 by γ-secretase

Six DNA constructs, each encoding an FAD-mutant γ-secretase, were generated, with all PSEN1 mutations corresponding to those under study by the Dominantly Inherited Alzheimer Network (DIAN). The selected mutations were S169L, S170F, G378E, F386S, A431E, A434T (*Figure 1B*). PSEN1 mutations S169L and S170F are in transmembrane domain 3 (TMD3), near the hinge region with TMD2. Mutations G378E and F386S are situated in TMD7, close to the catalytic aspartate residue D385 in the conserved GxGD motif. Mutations A431E and A434T are found in a highly conserved region that includes the PALP motif, which is essential for enzymatic activity (*Wang et al., 2004*; *Tomita et al., 2001*). The age of onset for these mutations varies as follows: S169L (29–31 years) (*Taddei et al., 1998*), S170F (mean 27 years) (*Snider et al., 2005*), G378E (38–44 years) (*Lanoiselée et al., 2017*), F386S (37–58 years) (*Raux et al., 2005*), A431E (mean 43 years) (*Dumois-Petersen et al., 2020*), and A434T (35 years) (*Jiao et al., 2014*). Monocistronic pMLINK plasmids encoding human PSEN1, each harboring one of these mutations, were individually prepared via mutagenesis of the wild-type (WT) construct. Each mutant PSEN1 DNA insert was cut out with restriction enzymes from

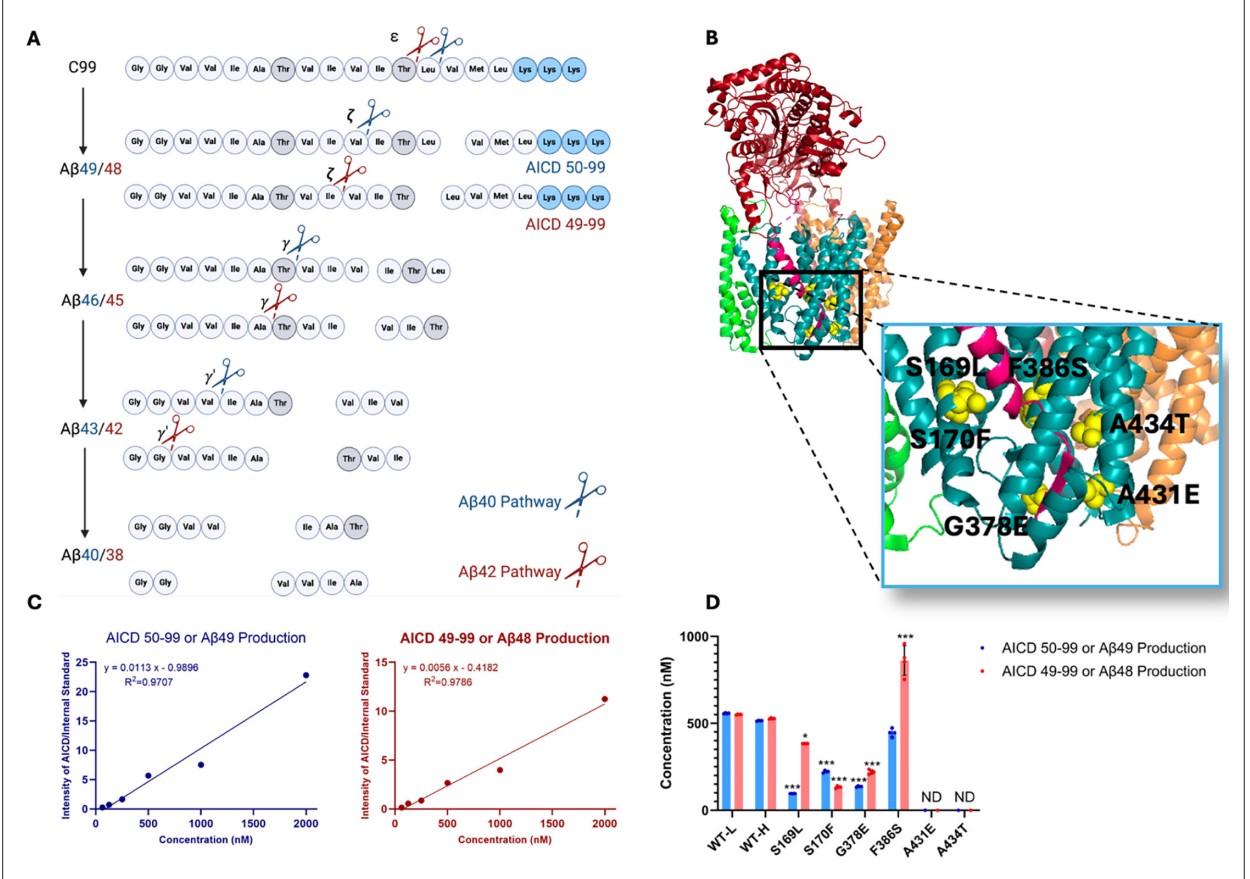

**Figure 1.** Effects of familial Alzheimer's disease (FAD)-mutant PSEN1 on endoproteolysis of C99 by γ-secretase. (**A**) Diagram of proteolytic cleavage of APP substrate by γ-secretase via two pathways. (**B**) Ribbon diagram of APP substrate bound to γ-secretase. The six FAD PSEN1 mutations studied in this paper are highlighted in yellow with side chain atoms as spheres. (**C**) Standard curves for APP intracellular domain (AICD) 50–99-Flag and AICD 49–99-Flag, coproducts of Aβ49 and Aβ48, respectively, were generated by matrix-assisted laser desorption/ionization time-of-flight (MALDI-TOF) using synthetic peptides, with insulin as an internal standard. (**D**) Quantification of AICD-Flag peptides from enzyme reactions of recombinant APP substrate C100-Flag with wild-type (WT) vs. FAD-mutant proteases by MALDI-TOF mass spectrometry (MS). Standard curves were used to quantify AICD levels for all reactions. Detection limits prevented measurement of AICD-Flag production below 62.5 nM (the lowest standard concentration) for two mutations (A431E and A434T). Consequently, these concentrations are marked as not determined (nd). In all graphs, n=4 and error bars represent s.d. Statistical comparisons between AICD product levels from FAD-mutant vs. wild-type (WT) enzymes were performed using unpaired two-tailed t-tests (*p<0.05, **p<0.01, ***p<0.001). All data describe biological replicates, and experiments were replicated in two independent experiments.

The online version of this article includes the following source data and figure supplement(s) for figure 1:

**Source data 1.** Spreadsheet contains mass spectrometry data for AICD49-99 and AICD50-99 associated with panels C and D.

**Figure supplement 1.** Process of installing PSEN-1 mutations into full γ-secretase plasmid.

**Figure supplement 2.** Expression, purification, and quality control of C100 substrate and γ-secretase.

**Figure supplement 2—source data 1.** Zip file contains images of western blots for gamma-secretase components.

**Figure supplement 2—source data 2.** Zip file contains a powerpoint file of western blots for gamma-secretase components, with labeling of molecular weight markers.

**Figure supplement 3.** Matrix-assisted laser desorption/ionization time-of-flight mass spectrometry (MALDI-TOF MS) detection of APP intracellular domain (AICD) 50–99 and AICD 49–99 products from wild-type (WT) and six PSEN1 familial Alzheimer's disease (FAD)-mutant γ-secretase.

this monocistronic plasmid and inserted into a tricistronic construct encoding the other three components of the γ-secretase complex (nicastrin, Aph-1, and Pen-2) through ligation-independent cloning (LIC). The resulting tetracistronic constructs encode all four components of the protease complex (*Figure 1—figure supplement 1*; *Lu et al., 2014*).

The tetracistronic constructs, each encoding either WT or one of the six FAD-mutant forms of the γ-secretase complex, were transiently transfected into human embryonic kidney (HEK)293F cells,

and the expressed protease complexes were subsequently purified (*Devkota et al., 2021*; *Devkota et al., 2024*). Concurrently, two versions of the recombinant FLAG epitope-tagged version of C99 substrate (C100-Flag) were expressed in *E. coli*: one in normal media, and the other in M9 minimal media containing $^{15}NH_4Cl$ and $^{13}C$-glucose as the sole sources of nitrogen and carbon, respectively. Purification provides light and heavy isotope-labeled C100-Flag (*Figure 1—figure supplement 2A*). Enzyme purity was verified by western blot analysis, which demonstrated the presence of all four components of γ-secretase, with PSEN1 autoproteolyzed into N-terminal and C-terminal fragments (NTF and CTF), indicative of maturation of the complex to the catalytically active protease (*Figure 1—figure supplement 2B*). Additionally, the quality of the substrates was assessed using matrix-assisted laser desorption/ionization time-of-flight mass spectrometry (MALDI-TOF MS) and silver staining (*Figure 1—figure supplement 2C*).

Each purified mutant protease (30 nM) was incubated with saturating levels of the heavy-isotope-labeled substrate (5 µM) at 37°C for 16 hr. In parallel, the WT enzyme was incubated with light-isotope-labeled C100-Flag substrate. Due to the slow rate for this reaction (WT enzyme $k_{cat}$ = 2 hr$^{-1}$), this long incubation period is still within the linear range for the formation of products (i.e. the enzyme remains saturated with substrate throughout the incubation period) (*Devkota et al., 2021*). The resulting products for all proteolytic events for each of these samples were then analyzed using MS techniques as described below. Due to the hydrophobicity and insolubility of the Aβ products, quantification by direct MS posed significant challenges. Consequently, we focused on analysis of the coproducts (AICDs and small peptides) and indirectly calculation of the concentration of the Aβ products.

Rates for the initial endoproteolytic step (ε cleavage) were determined by quantifying AICD species (*Figure 1A*) using MALDI-TOF MS. We utilized synthetic AICD 50–99-Flag and AICD 49–99-Flag peptides to construct standard concentration curves by MALDI-TOF MS (*Figure 1C*). This approach enabled accurate measurements of the production of AICD 50–99 (coproduct of Aβ49) and AICD 49–99 (coproduct of Aβ48) for both WT and mutant enzymes. For all standards and experimental samples, equal concentrations of insulin were added as an internal standard. The standard curves were generated based on the intensity ratio of the AICD parent ion to that of the internal standard (*Figure 1C*).

Using these standard curves, we calculated concentrations of AICD species in our test samples (*Figure 1D*). Equal concentrations of both AICD 50–99 and AICD 49–99 were produced when comparing WT protease incubated with light-isotope C100 to WT protease incubated with heavy-isotope C100. This demonstrates that use of heavy C100 does not affect the concentrations of products formed during the ε cleavage step. Quantification of AICD species was possible for four mutant proteases but not for A431E and A434T, as product concentrations from these two mutant enzymes were below the detection limit (*Figure 1D*, *Figure 1—figure supplement 3*), consistent with the known essential role of the PALP motif in proteolytic activity (*Wang et al., 2004*; *Tomita et al., 2001*). All FAD-mutant PSEN1-containing enzymes except for F386S exhibited a significant reduction in the production of both AICD 50–99 and AICD 49–99, which also measures formation of coproducts Aβ49 and Aβ48, respectively. Reduced ε cleavage of APP and Notch substrate has been observed with other PSEN1 FAD mutations (*Devkota et al., 2024*; *Song et al., 1999*; *Bentahir et al., 2006*). Uniquely, the PSEN1 F386S mutant protease resulted in a significant increase in production of AICD 49–99 and similar levels of AICD 50–99 compared to those observed with WT protease. Three other mutant enzymes (S169L, G378E, and F386S) likewise biased ε cleavage toward AICD 49–99, thereby favoring the Aβ48 → Aβ42 pathway, consistent with previous reports on PSEN1 FAD mutations (*Sato et al., 2003*).

Equal volumes of WT enzyme incubated with light-isotope C100 and PSEN1 FAD-mutant enzymes incubated separately with heavy-isotope C100 were combined in a 1:1 ratio after quenching the reactions (*Figure 2A*). The 1:1 mixtures were then used to quantify the coproducts of the trimming steps along the two canonical pathways Aβ49→Aβ46→Aβ43→Aβ40 and Aβ48→Aβ45→Aβ42→Aβ38 via liquid chromatography coupled with tandem MS (LC-MS/MS) (*Takami et al., 2009*; *Devkota et al., 2021*), as we previously described for six other FAD-mutant enzymes (*Devkota et al., 2024*). Mixing of the enzyme reactions after quenching but before analysis allows quantification of each small peptide coproduct formed from WT enzyme and mutant enzyme in the same LC-MS/MS run: Products from WT enzyme incubated with light-isotope C100 have lower mass than products from mutant enzyme incubated with heavy-isotope C100 (*Figure 2A*). Standard curves were generated for

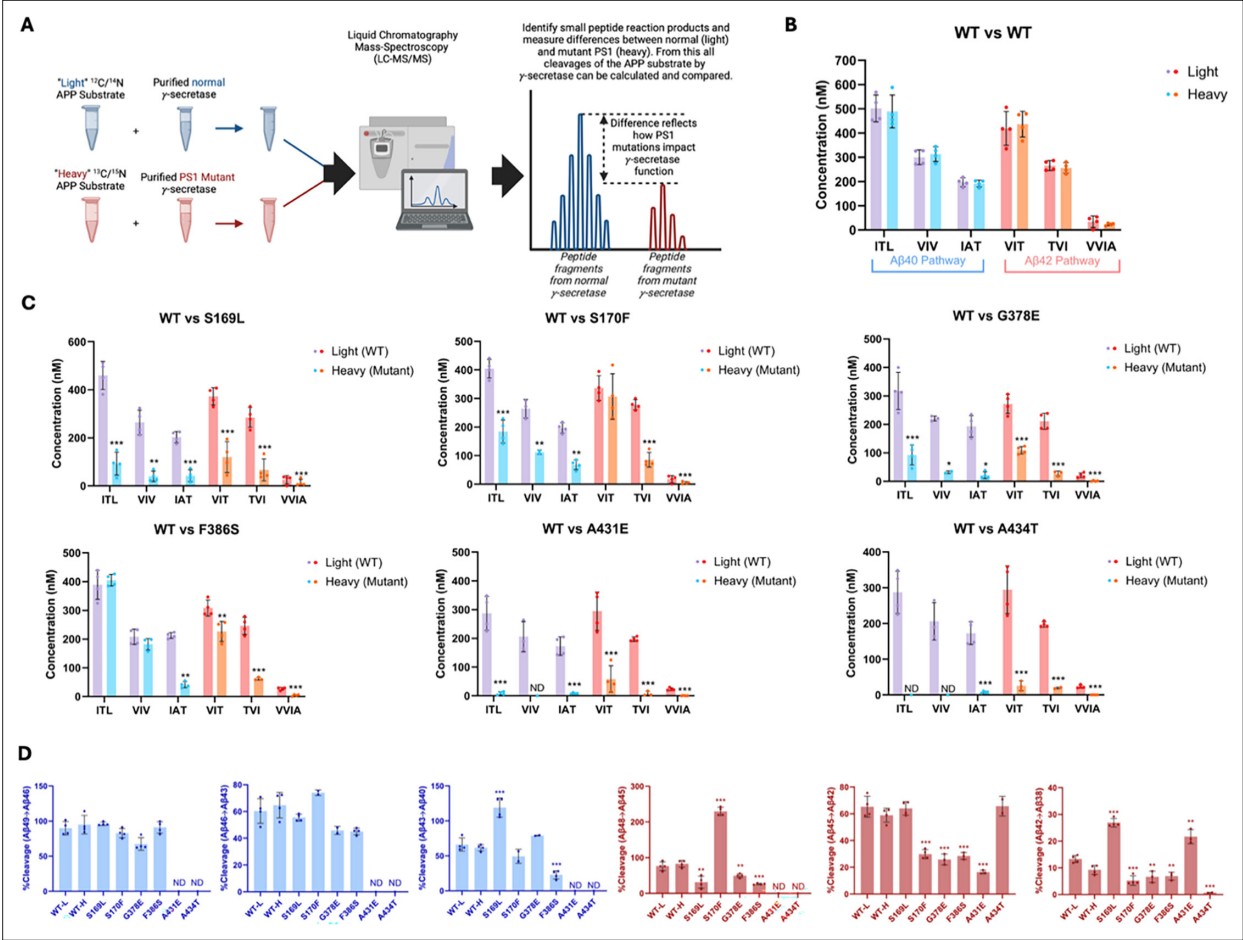

**Figure 2.** Alzheimer-mutant PSEN-1 affects the processive proteolysis of C99 by γ-secretase. (**A**) Schematic representation of the reaction mixtures and their preparation, analyzed by liquid chromatography coupled with tandem mass spectrometry (LC-MS/MS) for the detection of tri- and tetrapeptide coproducts. (**B**) Comparison of tri- and tetrapeptide coproduct concentrations between wild-type (WT) enzyme incubated with light-isotope substrate and WT enzyme incubated with heavy-isotope substrate, as analyzed by LC-MS/MS. (**C**) Bar graphs illustrating all coproduct formation for specific mutations. For the Aβ49→Aβ40 pathway, blue and purple bars represent the first, second, and third trimming steps. Red and orange bars denote trimming steps for the Aβ48→Aβ38 pathway. Purple and red bars indicate coproducts formed by WT γ-secretase, while blue and orange bars indicated coproducts formed by familial Alzheimer's disease (FAD)-mutant enzyme. (**D**) Bar graphs showing the percentage cleavage efficiency for each trimming step for mutant enzyme compared to WT enzyme. Cleavage events where the precursor Aβ peptide level was zero (i.e. no detected coproduct) are marked as not determined (nd). For each graph, n=4 and error bars represent s.d. Statistical significance was determined using unpaired two-tailed t-tests comparing FAD-mutant with WT enzyme reactions (*p<0.05, **p<0.01, ***p<0.001).

The online version of this article includes the following source data and figure supplement(s) for figure 2:

**Source data 1.** Spreadsheet contains mass spectrometry data for peptide products associated with panels B and C as well as calculations from these data associated with panel D.

**Figure supplement 1.** Alzheimer-mutant PSEN-1 affects the processive proteolysis of C99 by γ-secretase.

the tri- and tetrapeptide coproducts of each trimming step, using synthetic ITL, VIV, IAT, VIT, TVI, and VVIA peptides, allowing accurate determination of the concentrations of each coproduct generated from all WT and PSEN1 FAD-mutant enzyme reaction mixtures. Equal concentrations of each tri- and tetrapeptide were produced when comparing WT protease incubated with light-isotope C100 to WT protease incubated with heavy-isotope C100, demonstrating that the use of heavy-isotope C100 does not affect the concentrations of products formed during these cleavage steps (*Figure 2B*).

Quantification of small peptide coproducts from WT vs. PSEN1 FAD-mutant enzyme reactions revealed that each of the six mutations leads to a specific profile of effects on carboxypeptidase trimming. Mutations located within or near the conserved PALP motif (*Wang et al., 2004*; *Tomita et al., 2001*), A431E and A434T, exhibited significantly lower levels of trimming coproducts compared to WT

and other mutants (*Figure 2C*, *Figure 2—figure supplement 1*). Although AICD levels from A431E and A434T PSEN1 mutant proteases were below those of the lowest standards, very low levels of certain small peptide coproducts were detected within the range of their standard curves. In strong contrast, F386S PSEN1 mutant enzyme produced equal levels of ITL (coproduct of Aβ49→Aβ46) and VIV (coproduct of Aβ46→Aβ43) compared to WT. However, levels of IAT (coproduct of Aβ43→Aβ40) were substantially lower for this mutant enzyme, indicating a buildup of Aβ43 (*Figure 2C*, *Figure 2—figure supplement 1*). Along the Aβ42 pathway, F386S PSEN1 enzyme produced levels of VIT (coproduct of Aβ48→Aβ45) only slightly lower than WT enzyme; however, because F386S generated increased levels of AICD 49–99 (coproduct of C99→Aβ48), this mutant also led to a buildup Aβ48. For mutants S169L, S170F, and G378E, small peptide coproducts were generally lower than those generated from WT (*Figure 2C*, *Figure 2—figure supplement 1*), consistent with their reduced initial ε proteolysis (*Figure 1D*).

Quantification of all coproducts (AICDs, small peptides) using MS techniques enabled calculation of the percent cleavage for each trimming step in the processive proteolysis of APP by γ-secretase (*Figure 2D*). That is, given the level of Aβ precursor produced, how much of this precursor was cleaved in the next step? This analysis revealed that none of the mutant enzymes exhibited deficiencies in the Aβ49→Aβ46 or Aβ46→Aβ43 cleavage steps compared to WT enzyme, although percent cleavage could not be determined for A431E and A434T, as these mutant enzymes produced coproducts AICD 50–99 and ITL below the limits of detection. The S169L and F386S mutations demonstrated significant changes in the Aβ43→Aβ40 step, with S169L showing a marked increase and F386S showing a decrease compared to WT. All mutations were deficient in the Aβ48→Aβ45 cleavage step, except for S170F, which exhibited an increase in this trimming step. It should be pointed out, however, that this increased percent efficiency of S170F—over 200%—is not possible, suggesting either underdetection of AICD 49–99 (production of Aβ48) or over-detection of VIT (degradation of Aβ48) for this mutant enzyme. The reason for this discrepancy is not clear. All mutations except for S169L and A434T were also deficient in the Aβ45→Aβ42 trimming step. Furthermore, all mutations were deficient in the Aβ42 →Aβ38 cleavage process, except for S169L and A431E, which showed increases compared to WT enzyme. We should point out, however, that detected products from A431E and A434T were very low and therefore likely making calculations of percent cleavage unreliable. This detailed analysis highlights specific deficiencies and alterations in the cleavage patterns associated with each mutation, providing a comprehensive and quantitative understanding of how these mutations affect the proteolytic processing of APP by γ-secretase. Overall, calculation of percent efficiency for each trimming step, along with analysis of effects on initial ε cleavage, revealed that all PSEN1 FAD-mutant proteases are deficient in multiple cleavage steps in processive proteolysis of APP substrate, consistent with the previous findings for the other six PSEN1 FAD mutations (*Devkota et al., 2024*).

**Table 1.** Calculated concentration (nM) of each Aβ variant resulting from processing APP substrate by wild-type (WT) vs. familial Alzheimer's disease (FAD)-mutant $\gamma$ -secretase*.

| PSEN-1 | Aβ49 | Aβ46 | Aβ43 | Aβ40 | Aβ48 | Aβ45 | Aβ42 | Aβ38 |
|--------|------|------|------|------|------|------|------|------|
| WT-L | 56.0 | 201.7 | 103.3 | 196.9 | 131.5 | 153.3 | 231.6 | 34.2 |
| WT-H | 26.3 | 176.3 | 121.0 | 191.8 | 91.2 | 180.9 | 232.0 | 23.7 |
| S169L | 3.7 | 52.7 | –2.8 | 42.5 | 264.6 | 52.8 | 53.8 | 12.9 |
| S170F | 38.4 | 128.3 | –2.8 | 67.9 | –173.5 | 221.2 | 80.8 | 4.9 |
| G378E | 44.2 | 76.7 | –5.5 | 21.6 | 112.2 | 88.9 | 19.2 | 1.3 |
| F386S | 41.4 | 222.9 | 140.9 | 41.2 | 634.8 | 180.1 | 43.3 | 4.3 |
| A431E | ND | 8.3 | –8.8 | 8.8 | ND | 52.6 | 5.1 | 0.4 |
| A434T | ND | 0 | –7.9 | 7.9 | ND | 9.2 | 9.4 | 0.3 |

*Calculated Aβ species produced from reaction mixtures containing 5 µM C100-Flag incubated at 37°C for 16 hr with 30 nM of either WT or FAD-mutant γ-secretase complexes. Calculations were based on concentrations of coproducts, where [Aβx] = [coproduct of Aβx production] – [coproduct of Aβx degradation] (e.g. [Aβ48] = [APP intracellular domain [AICD] 49–99] – [VIT]).

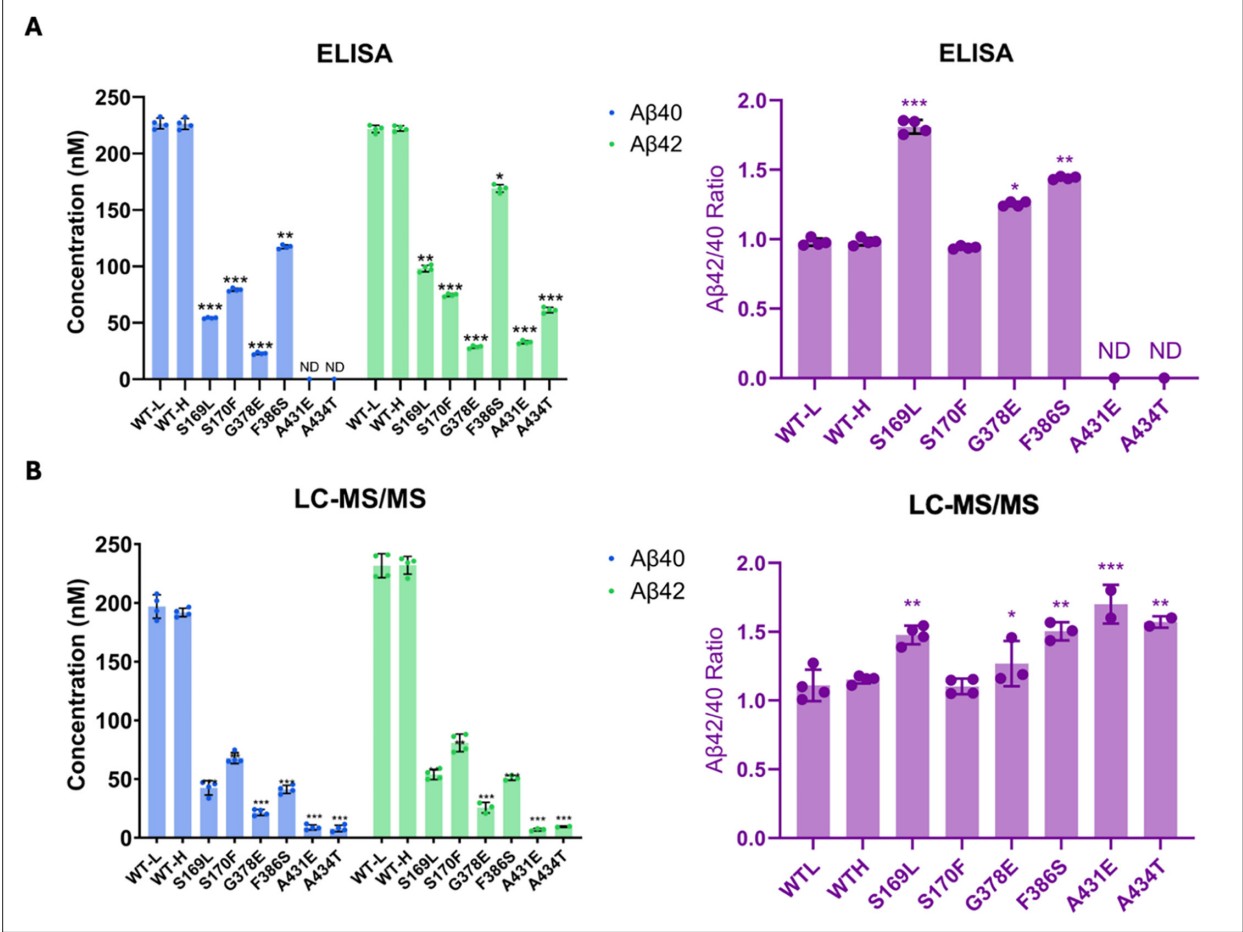

**Figure 3.** Comparison of Aβ40 and Aβ42 concentrations determined by liquid chromatography coupled with tandem mass spectrometry (LC-MS/MS) vs. ELISA. Final Aβ40 and Aβ42 concentrations upon incubation of purified γ-secretase with C100-Flag and the resulting Aβ42/Aβ40 ratios assessed by (**A**) ELISAs and (**B**) LC-MS/MS calculations. n=4; error bars represent s.d.; unpaired two-tailed t-tests comparing familial Alzheimer's disease (FAD)-mutant to wild-type (WT) enzyme reactions (*p≤0.05, **p≤0.01, ***p≤0.001). All data describe biological replicates, and experiments were replicated in two independent experiments.

The online version of this article includes the following source data for figure 3:

**Source data 1.** Spreadsheet contains ELISA data for Abeta40 and Abeta42 associated with panel A.

By analyzing the extent of degradation and production of each Aβ peptide, we were able to calculate the levels of each Aβ peptide in the final quenched enzyme reaction mixtures (***Table 1***). We observed a negative value for the concentration of Aβ48 produced from the S170F enzyme, due to the discrepancy pointed out above in quantifying the coproducts of Aβ48 production and degradation. To validate the calculated concentrations of Aβ peptides obtained by LC-MS/MS, we used Aβ40- and Aβ42-specific ELISAs (***Figure 3A***). The ELISA results were consistent with the calculations from the LC-MS/MS data for all mutations except for F386S. This mutation exhibited higher concentrations of both Aβ40 and Aβ42 when measured by ELISA compared to the calculations derived from MS data (***Figure 3A vs. B***). Because the PSEN1 F386S mutant enzyme produced high levels of Aβ43 (***Table 1***), as previously reported (***Liu et al., 2023***), we tested for cross-reactivity in the ELISAs with synthetic Aβ43 peptide. We found that the putatively Aβ42- and Aβ40-specific ELISAs both cross-reacted with Aβ43 (***Supplementary file 1***, ***Supplementary file 2***), which can explain the observed discrepancies with F386S enzyme. According to the ELISA results, all six PSEN1 FAD-mutant enzymes produced lower levels of Aβ40 and Aβ42 compared to WT, consistent with our previous findings with six other PSEN1 FAD mutations (***Devkota et al., 2024***). Mutants S169L, G378E, and F386S showed an increased Aβ42/Aβ40 ratio by both ELISA and LC-MS/MS (***Figure 3***). In contrast, Aβ42/Aβ40 produced from the S170F mutant enzyme was equivalent to that of WT by both methods. Mutants A431E and A434T appear to

increase Aβ42/Aβ40 by LC-MS/MS; however, these two FAD-mutant enzymes produce such low levels of peptide coproducts that the calculated ratio is not reliable. By ELISA, only Aβ42 was detectable from A431E and A434T mutant enzymes, precluding calculation of Aβ42/Aβ40 ratios.

All data describe biological replicates, and experiments were replicated in two independent experiments.

## PSEN1 FAD mutations stabilize γ-secretase E-S complexes

To test the effects of FAD mutations on the stability of γ-secretase E-S complexes, we conducted FLIM in intact cells (*Elangovan et al., 2002*). WT or FAD-mutant PSEN1 was co-expressed with C99 substrate in HEK293 cells in which endogenous PSEN1 and PSEN2 were knocked out through CRISPR/Cas9 gene editing (*Liu et al., 2021*). The C99 substrate construct consisted of human APP C99 flanked by an N-terminal signal peptide and a C-terminal near-infrared fluorescence protein: miRFP720 (C99-720) (*Figure 4A*). Transfected cells were fixed and permeabilized and treated with primary antibodies that bind to the N-terminal region of C99/Aβ (mouse antibody 6E10) and with an epitope on the nicastrin component of γ-secretase (rabbit antibody NBP2-57365) that lies in close proximity to the N-terminal region of bound C99/Aβ. Secondary antibodies conjugated to fluorophores (anti-mouse IgG antibody conjugated with Alexa Fluor 488 and anti-rabbit IgG antibody conjugated with Cy3) were then added. In this experimental design, reduction in the fluorescence lifetime of the Alexa Fluor 488, through fluorescence resonance energy transfer to Cy3, indicates E-S complex detection. Importantly, use of rabbit antibody toward a distal region in the γ-secretase complex results in no change in Alexa Fluor 488 fluorescence lifetime (*Devkota et al., 2024*).

For each but one PSEN1 FAD mutant tested, Alexa Fluor 488 fluorescence lifetime is reduced compared to that of WT PSEN1 (*Figure 4B and C*), consistent with increased stability of E-S complexes. The miRFP720 fusion to the C-terminus of C99 provides a means of distinguishing between C99-rich regions in the cells (low ratio of 6E10 to C99-720) from Aβ-rich regions (high ratio of 6E10 to C99-720) (*Figure 4A*; *Devkota et al., 2024*; *Maesako et al., 2022*; *McKendell et al., 2022*). Although the PSEN1 F386S mutant did not show an overall fluorescence lifetime reduction compared to WT, when differentiating C99-rich regions vs. Aβ-rich regions (*Figure 4D*), fluorescence lifetime was significantly reduced in Aβ-rich regions and not C99-rich regions (*Figure 4E vs. F*). This finding is consistent with the results of proteolytic analysis, as PSEN1 F386S is the only mutant that was not deficient in ε cleavage but was deficient in specific trimming steps, especially Aβ43→Aβ40 and Aβ48→Aβ45. We should point out that the observed decrease in donor lifetime with the PSEN1 FAD mutants might also be due to lower levels of C99-720 expression or higher levels of PSEN1 CTF (i.e. mature γ-secretase complexes). However, C99-720 intensities are not significantly different between cells transfected with WT and those with FAD PSEN1 (*Figure 4—figure supplement 1A*). Furthermore, western blot analysis shows that levels of C99-720 are not significantly low and those of PSEN1 CTF are not high in FAD PSEN1 compared to WT PSEN1 expressing cells (*Figure 4—figure supplement 1B and C*). Although PSEN1 CTF levels trend low for PSEN1 F386S, this mutant resulted in decreased FLIM only in Aβ-rich regions. Thus, the reduced Alexa 488 fluorescence lifetime apparently reflects effects of FAD mutation on E-S complex stability.

## Discussion

The persistent challenge of developing effective therapeutics for AD continues to spark contentious debates within the biomedical research community on the identity of pathogenic triggers. The 'amyloid cascade hypothesis,' long a central focus in AD research, has faced significant scrutiny, particularly due to its failure to consistently correlate brain amyloid plaque burden with cognitive decline (*Herrup, 2015*). Difficulties in pinpointing disease drivers of AD and in discovering effective therapeutics suggest that entities and processes beyond Aβ might play pivotal roles in initiating neurodegeneration. Focusing on FAD could simplify identification of pathogenic mechanisms, as these rare variants of AD are caused by dominant missense mutations in the substrate and enzyme that produce Aβ. This targeted approach may provide clearer insights into the molecular underpinnings of AD and facilitate the development of more effective treatments.

In this study, we focused on six FAD PSEN1 mutations that are among those under investigation by the DIAN (*Storandt et al., 2014*; *McKay et al., 2023*; *Schultz et al., 2024*), conducting a

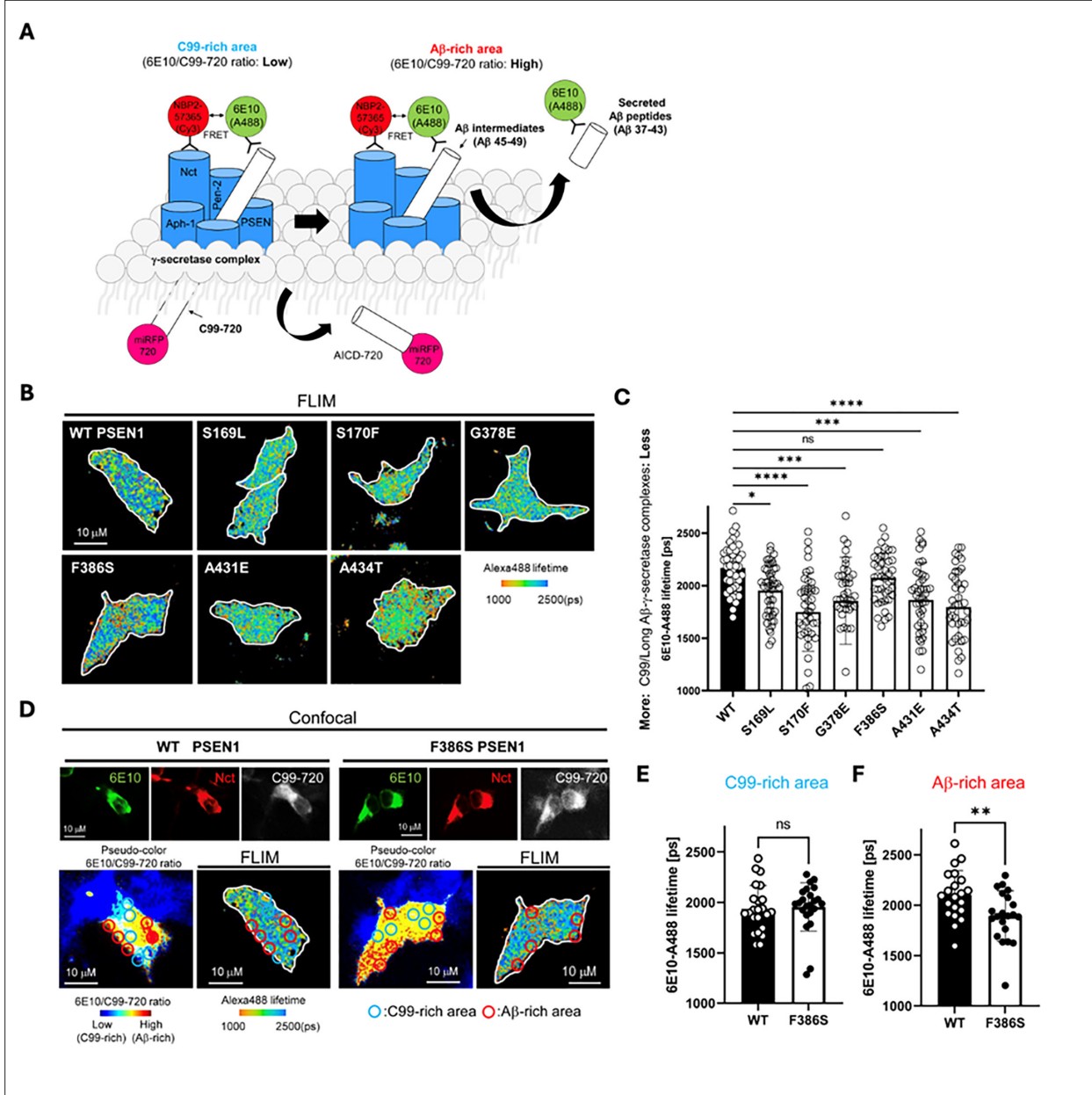

**Figure 4.** Familial Alzheimer's disease (FAD)-mutant PS1 stabilizes γ-secretase enzyme-substrate (E-S) interaction. (**A**) Design of fluorescence lifetime imaging microscopy (FLIM) set-up to detect E-S complexes of γ-secretase and C99/Aβ intermediates. 6E10-Alexa Fluor 488 over C99-720 fluorescence ratio (6E10-A488/C99-720 ratio) enables distinguishing C99-rich and Aβ-rich subcellular compartments. (**B**) PSEN1/2 dKO human embryonic kidney (HEK)293 cells were co-transfected with C99-720 and wild-type (WT) or FAD-mutant PSEN1. Transfected cells were immunostained with anti-C99/Aβ (mouse 6E10) and anti-nicastrin (rabbit NBP2-57365) primary antibodies and Alexa Fluor 488 (FRET donor) or Cy3 (acceptor)-conjugated anti-mouse and anti-rabbit IgG secondary antibodies, respectively. The donor 6E10-Alexa Fluor 488 (6E10-A488) lifetime was measured by FLIM. Energy transfer from the donor to the acceptor results in shortening of the donor lifetime. Scale bars, 10 μm. (**C**) 6E10-A488 lifetimes were analyzed in randomly selected regions of interest (ROIs) (n=40–47 from 6 to 8 cells), highlighting increased E-S complexes in the cells with FAD PSEN1 mutants, except F386S, compared to WT controls. One-way ANOVA and Tukey's multiple comparisons test; n.s., p>0.05; *p<0.05; **p<0.01; ***p<0.001; ****p<0.0001. (**D**) Representative images of confocal, pseudo-color analysis to identify C99 or Aβ-rich subcellular areas and corresponding FLIM in WT or F386S PSEN1 expressing cells. Scale bars, 10 μm. (**E**) In the areas with lower 6E10-A488/C99-720 ratios (i.e. C99-rich areas), 6E10-A488 lifetimes were not different between the cells with WT PSEN1 and those with F386S mutant. n=21 ROIs. (**F**) On the other hand, 6E10-A488 lifetimes were significantly shorter in the cells expressing F386S mutant PSEN1 compared to WT controls in Aβ-rich ROIs (n=23). Unpaired t-test **p<0.01. All data describe biological replicates, and experiments were replicated in three independent experiments. Error bars represent s.d. in all cases.

The online version of this article includes the following source data and figure supplement(s) for figure 4:

*Figure 4 continued on next page*

*Figure 4 continued*

**Source data 1.** Spreadsheet contains fluorescence lifetime data associated with panels C, E and F.

**Figure supplement 1.** Protein expression of human embryonic kidney (HEK)293 cells cotransfected with C99-720 and PSEN1 variants.

**Figure supplement 1—source data 1.** Spreadsheet contains fluorescence lifetime data associated with panel A.

**Figure supplement 1—source data 2.** Original images of western blots for panel B.

**Figure supplement 1—source data 3.** Powerpoint file of images of western blots for panel B with labeling.

comprehensive and quantitative analysis of processive proteolysis of APP substrate C99 by γ-secretase. Our analysis revealed that five of the six mutations resulted in substantial deficiencies in the initial cleavage event (ε cleavage), consistent with our previously reported findings for six other FAD PSEN1 mutations (*Devkota et al., 2024*). The exception was the F386S mutation, which was deficient in several trimming steps (Aβ43→Aβ40, Aβ48→Aβ45, Aβ45→Aβ42, and Aβ42→Aβ38) but not in ε cleavage. This was surprising, because this mutation is immediately adjacent to one of the two catalytic aspartate residues (D385) and was expected to decrease ε cleavage.

To further explore the impact of these mutations, we employed FLIM. Stabilization of FAD-mutant E-S complexes while stalled in their proteolytic activities was supported by reduced fluorescence lifetimes of labeled antibody probe combinations targeting the E-S complexes. Based on our FLIM studies, all six FAD PSEN1 mutations tested showed stabilized E-S complexes. Five of these mutations stabilized overall γ-secretase E-S interactions, while the F386S mutant stabilized only γ-secretase/Aβ and not γ-secretase/C99 interactions. Together, FLIM data and comprehensive analysis of all cleavage

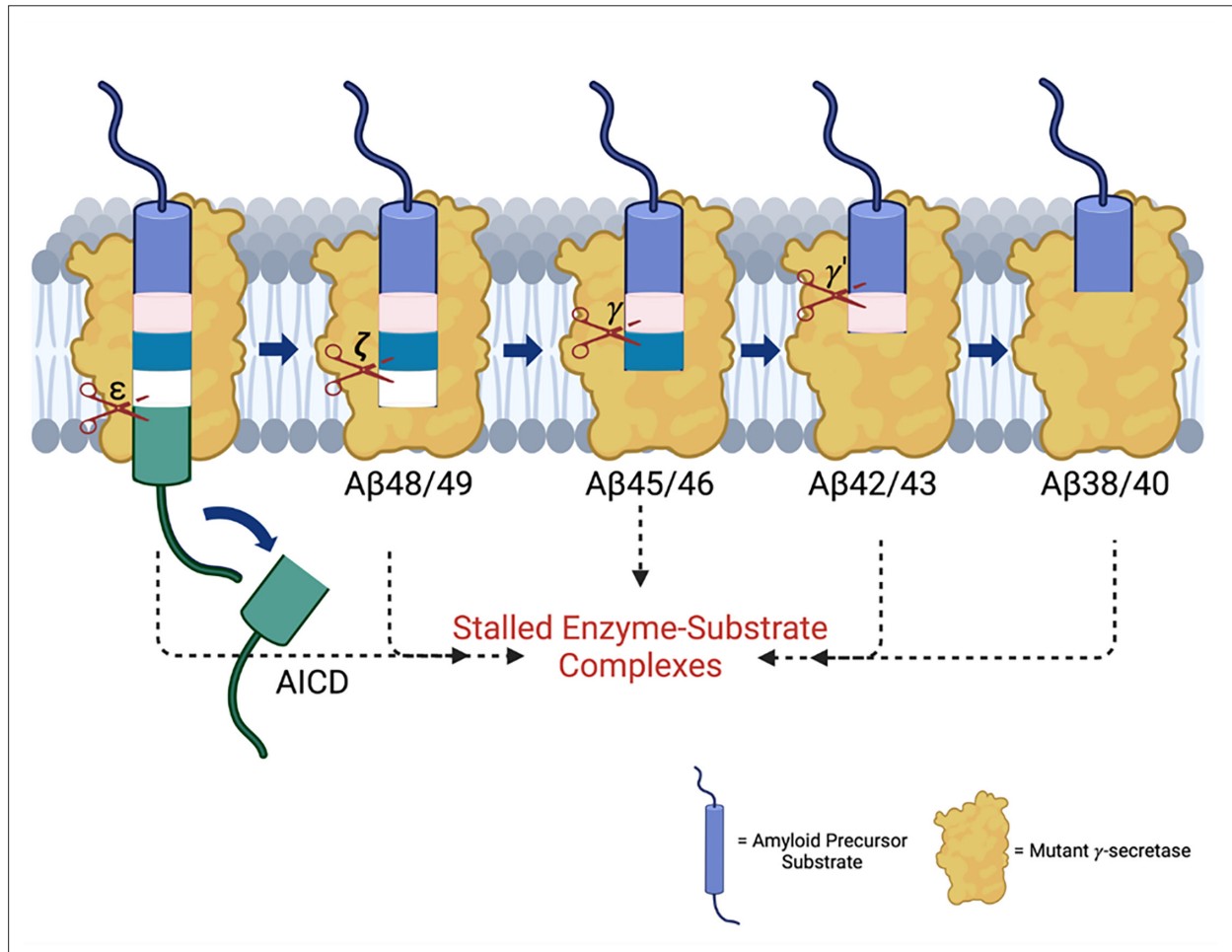

**Figure 5.** Familial Alzheimer's disease (FAD) mutations lead to stalled processive proteolysis of APP substrate by γ-secretase. PSEN1 FAD-mutant γ-secretase complexes are deficient in specific cleavage steps. Enzyme-substrate/intermediate complexes are stalled at these stages.

steps involved in the processive proteolysis of APP substrate by γ-secretase revealed that each of these mutations is deficient in multiple cleavage events due to stalled and stabilized E-S complexes (*Figure 5*).

Among the six mutants, the FAD S169L mutant was the second least deficient in ε cleavage, as demonstrated by quantification of AICDs using MALDI-TOF. Similarly, FLIM studies indicated this mutation results in the smallest decrease in overall fluorescence lifetime after F386S, suggesting a small but still significant increase in stabilization of overall E-S complexes. According to recent reports by Guo et al., and Odorčić et al. of cryoelectron microscopy (cryoEM) structures of γ-secretase bound to C99 and Aβ intermediates, S169 is a key residue involved in processive proteolysis. These two groups suggested that FAD mutants destabilize E-S complexes, specifically mentioning the loss of an H-bonding interaction with S169 mutants as the rationale (*Guo et al., 2024*; *Odorčić et al., 2024*). However, static structures such as those captured by cryoEM cannot account for dynamic changes in protein conformation, such as the decreased E-S complex flexibility we observed for FAD mutations by molecular dynamics simulations (*Devkota et al., 2024*).

The PSEN1 A431E and A434T FAD-mutant enzymes exhibited the most pronounced deficiencies in all cleavage events compared to WT. These mutations are located within or near the highly conserved PALP motif, which plays a critical role in the active site conformation and catalytic activity of γ-secretase (*Wang et al., 2004*; *Tomita et al., 2001*). This motif is essential for the recognition of APP substrate by γ-secretase (*Zhou et al., 2019*). The C-terminal PALP motif and the rest of PSEN1 TMD9 are also integral to the formation of the catalytic pore of the protease (*Sato et al., 2008*). Consequently, the observed deficiencies in A431E and A434T can be attributed to the disruption of these crucial structural and functional elements. Nevertheless, our FLIM results suggest these FAD-mutant proteases can still interact with C99 and form stable E-S complexes. In this context, it should be noted that no FAD-mutant enzyme is completely deficient in proteolytic activity; substrate interaction with the mutant protease is apparently critical to pathogenesis. Consistent with this idea, true loss of function of γ-secretase—due to dominant mutations in PSEN1 and other components of the protease complex that lead to nonsense-mediated decay (i.e. haploinsufficiency)—cause a hereditary skin disease, not neurodegeneration (*Wang et al., 2010*).

While an elevated Aβ42/Aβ40 ratio is commonly cited as a hallmark of FAD mutations, our results show that S170F did not lead to an increase in this ratio. This finding is consistent with a previous study, which analyzed Aβ40 and Aβ42 production from 138 FAD PSEN1 mutations using purified γ-secretase and APP substrate, which revealed that many FAD mutations do not elevate Aβ42/Aβ40 (*Sun et al., 2017*). Among those mutations that do increase the ratio, the effect is primarily due to a reduction in Aβ40 production, with some mutations resulting in minimal production of both Aβ variants.

The findings reported herein are consistent with our working hypothesis that FAD mutations lead to stalled γ-secretase E-S complexes that contribute to pathogenesis (*Figure 5*). These stalled complexes—observed in FAD regardless of the specific deficient cleavage steps of the mutation—can trigger synaptic loss in vivo, even in the absence of Aβ production (*Devkota et al., 2024*). The 'stalled complex hypothesis' posits that stabilized E-S complexes, even in the absence of Aβ42 or any other proteolytic product, can initiate pathogenesis. In this context, γ-secretase activators that rescue stalled E-S complexes offer a promising therapeutic strategy by potentially rescuing deficient proteolytic function and thereby reducing levels of stalled E-S complexes, without over-activating cleavage of other substrates (which are limited by prior rate-determining ectodomain shedding) (*Wolfe, 2024*). Such activators would be distinct from γ-secretase modulators, which selectively reduce Aβ42 levels by stimulating Aβ42→Aβ38. This approach may complement therapies targeting tau aggregation, lipid metabolism, and other Alzheimer-associated pathways, potentially synergizing with emerging drug candidates (*Cummings et al., 2023*).

## Limitations of the study

While detailed and quantitative studies of the impact of FAD mutations on all proteolytic stages of γ-secretase processing of C99 have been carried out here, these experiments utilized purified enzymes and substrates in a detergent-solubilized system. This system differs significantly from the cell membrane environment, which is characterized by complex lipid and protein compositions and dynamic microdomains that can affect protein folding, stability, and function. Therefore, the effects of these mutations on γ-secretase processing of C99 may vary considerably when studied in

detergent-solubilized systems compared to their natural membrane environment. However, we previously noted that detergent-solubilized and reconstituted proteoliposome systems give similar ratios of AICD 49–99/AICD 50–99 and Aβ42/Aβ40 (*Devkota et al., 2021*). While using MALDI-TOF instead of western blotting provided more accurate quantification of AICDs, we encountered challenges with lower concentrations of AICDs produced by mutations such as A431E and A434T. The concentrations of AICDs in these mostly deficient mutations fell below the detection limit of our instrument, limiting our ability to assess their levels accurately. In addition, FLIM studies relied on overexpression of PSEN1 and C99-720, which could lead to artifacts.

# Materials and methods

## Key resources table

| Reagent type (species) or resource | Designation | Source or reference | Identifiers | Additional information |
|---|---|---|---|---|
| Antibody | Mouse anti-Flag M2 | Sigma-Aldrich | Cat. No. F1804; RRID:Ab_262044 | |
| Antibody | Mouse anti-FLAG M2-agarose beads | Sigma-Aldrich | Cat. No. A2220; RRID:AB_10063035 | |
| Antibody | Mouse anti-PSEN1-NTF | Bio-Legend | Cat. No. 823401; RRID:AB_2564868 | |
| Antibody | Rabbit anti-nicastrin | Novus Biologicals | Cat. No. NBP2-57365 | |
| Antibody | Presenilin 1 (D39D1) Rabbit mAb-CTF | Cell Signaling | Cat. No. 5643 | |
| Antibody | Rabbit anti-Aph-1a, 245–265 (C-terminus) Antibody | Bio-Legend | Cat. No. 823101 | |
| Antibody | Goat anti-Mouse IgG (H+L) Secondary Antibody, HRP | Invitrogen | Cat. No. 62–6520 | |
| Antibody | Anti-rabbit IgG, HRP-linked Antibody | Cell Signaling | Cat. No. 7074 | |
| Antibody | Precision Plus Protein WesternC Blotting Standards | Bio-Rad | Cat. No. 1610376 | |
| Antibody | Precision Protein StrepTactin-HRP Conjugate | Bio-Rad | Cat. No. 1610381 | |
| Strain, strain background (*Escherichia coli*) | NEB 5-alpha Competent *E. coli* (DH5α) | New England Biolabs | Cat. No. C2987H | |
| Strain, strain background (*Escherichia coli*) | *E. coli* BL21 DE3 | New England Biolabs | Cat. No. C2530H | |
| Chemical compound, drug | Expi293 Expression Medium | Thermo Fisher Scientific | Cat. No. A1435101 | |
| Chemical compound, drug | $^{13}C$ glucose | Cambridge Isotope Laboratories | Cat. No. CLM-1396 | |
| Chemical compound, drug | $^{15}NH_4Cl$ | Cambridge Isotope Laboratories | Cat. No. NLM-467 | |
| Chemical compound, drug | PacI restriction enzyme | New England Biolabs | Cat. No. R0547S | |
| Chemical compound, drug | SwaI restriction enzyme | New England Biolabs | Cat. No. R0604S | |
| Chemical compound, drug | T4 DNA Polymerase | New England Biolabs | Cat. No. M0203L | |
| Chemical compound, drug | DOPC (1,2-dioleoyl-*sn*-glycerol-3-phosphocholine) | Avanti Polar Lipids | Cat. No. 850375 | |
| Chemical compound, drug | DOPE (1,2-dioleoyl-*sn*-glycero-3-phosphoethanolamine) | Avanti Polar Lipids | Cat. No. 850725 | |
| Chemical compound, drug | Digitonin | Goldbio | Cat. No. D-180–5 | |
| Chemical compound, drug | Opti-MEM I Reduced Serum Medium | Gibco | Cat. No. 31985062 | |
| Chemical compound, drug | SuperSignal West Femto Maximum Sensitivity Substrate | Thermo Fisher Scientific | Cat. No. 34094 | |
| Chemical compound, drug | ExpiFectamine 293 Transfection Kit | Gibco | Cat. No. A14524 | |
| Chemical compound, drug | LB Broth | Fisher Scientific | Cat. No. BP1426-2 | |
| Chemical compound, drug | UltraPure Ethidium Bromide | Invitrogen | Cat. No. 15585011 | |
| Chemical compound, drug | MES SDS Running Buffer | Invitrogen | Cat. No. B0002 | |
| Chemical compound, drug | Western Blot Stripping Buffer | Thermo Fisher Scientific | Cat. No. 21059 | |
| Chemical compound, drug | NuPAGE Transfer Buffer | Invitrogen | Cat. No. NP00061 | |

*Continued on next page*

*Continued*

| Reagent type (species) or resource | Designation | Source or reference | Identifiers | Additional information |
|---|---|---|---|---|
| Chemical compound, drug | TAE Buffer, Molecular Biology Grade (Tris-acetate-EDTA) | Promega | Cat. No. V4271 | |
| Chemical compound, drug | SimplyBlue SafeStain | Invitrogen | Cat. No. LC6060 | |
| Chemical compound, drug | NuPAGE Bis-Tris Mini Protein Gels, 10%, 1.0–1.5 mm | Invitrogen | Cat. No. NP0315BOX | |
| Chemical compound, drug | NuPAGE LDS Sample Buffer (4×) | Invitrogen | Cat. No. NP0007 | |
| Chemical compound, drug | NuPAGE Sample Reducing Agent (10×) | Invitrogen | Cat. No. NP0009 | |
| Chemical compound, drug | EDTA (0.5 M), pH 8.0 | Thermo Fisher Scientific | Cat. No. R1021 | |
| Chemical compound, drug | TriTrack DNA Loading Dye (6×) | Thermo Fisher Scientific | Cat. No. R1161 | |
| Chemical compound, drug | Penicillin-Streptomycin | Gibco | Cat. No. 15140122 | |
| Peptides, recombinant proteins | Peptides VIT, ITL, VIV, TVI, IAT, and VVIAA | New England Peptide | Custom synthesized | |
| Peptides, recombinant proteins | beta-Amyloid Peptide (1–43) (Aß43) | Abcam | Cat. No. 134500-80-4 | |
| Peptides, recombinant proteins | AICD 50–99 | Chemical Biology Synthetic Core at The University of Kansas | N/A | |
| Peptides, recombinant proteins | AICD 49–99 | Chemical Biology Synthetic Core at The University of Kansas | N/A | |
| Peptides, recombinant proteins | ProteoMass Insulin MALDI-MS | Sigma-Aldrich | Cat. No. 11070-73-8 | |
| Commercial assay or kit | QuikChange Lightning Multi-Site Directed Mutagenesis kit | Agilent | Cat. No. 210513 | |
| Commercial assay or kit | Amyloid b-peptide 1–40 ELISA kit | Invitrogen | Cat. No. KHB3481 | |
| Commercial assay or kit | Amyloid b-peptide 1–42 ELISA kit | Invitrogen | Cat. No. KHB3441 | |
| Commercial assay or kit | Pierce BCA Protein Assay Kits | Thermo Fisher Scientific | Cat. No. 23225 | |
| Commercial assay or kit | PureLink HiPure Plasmid Filter Maxiprep Kit | Invitrogen | Cat. No. K210017 | |
| Commercial assay or kit | QIAwave Plasmid Miniprep Kit | QIAGEN | Cat. No. 27204 | |
| Commercial assay or kit | QIAquick Gel Extraction Kit | QIAGEN | Cat. No. 28704 | |
| Cell line (human embryonic kidney cells) | Expi293F cells | Thermo Fisher Scientific | Cat. No. A14527 | |
| Sequence-based reagent | DNA primers for S169L PSEN1 mutagenesis | Invitrogen | Custom synthesized | catgcctggcttattatattatctctattgttgctgttc |
| Sequence-based reagent | DNA primers for S170F PSEN1 mutagenesis | Invitrogen | Custom synthesized | catgcctggcttattatatcatttctattgttgctgttc |
| Sequence-based reagent | DNA primers for G378E PSEN1 mutagenesis | Invitrogen | Custom synthesized | gacccagaggaaagggaagtaaaacttggattg |
| Sequence-based reagent | DNA primers for F386S PSEN1 mutagenesis | Invitrogen | Custom synthesized | cttggattgggagattccattttctacagtgttctg |
| Sequence-based reagent | DNA primers for A431E PSEN1 mutagenesis | Invitrogen | Custom synthesized | gccattttcaagaaagaattgccagctcttccaatc |
| Sequence-based reagent | DNA primers for A434T PSEN1 mutagenesis | Invitrogen | Custom synthesized | aagaaagcattgccaactcttccaatctccatc |
| Recombinant DNA reagent | pMLINK-PSEN1 | Coauthor Y Shi | *Lu et al., 2014* | |
| Recombinant DNA reagent | pMLINK-Aph1 (with C-terminal HA epitope tag) | Coauthor Y Shi | *Lu et al., 2014* | |
| Recombinant DNA reagent | pMLINK-NCT (with C-terminal V5 and 6XHIS epitope tags) | Coauthor Y Shi | *Lu et al., 2014* | |
| Recombinant DNA reagent | pMLINK-Pen-2 (with N-terminal STREP and FLAG epitope tags) | Coauthor Y Shi | *Lu et al., 2014* | |
| Recombinant DNA reagent | pMLINK-PSEN1-Aph1-NCT-Pen-2 | This study | *Lu et al., 2014* | |
| Recombinant DNA reagent | pMLINK-PSEN1(S169L)-Aph1-NCT-Pen-2 | This study | N/A | |
| Recombinant DNA reagent | pMLINK-PSEN1(S170F)-Aph1-NCT-Pen-2 | This study | N/A | |
| Recombinant DNA reagent | pMLINK-PSEN1(G378E)-Aph1-NCT-Pen-2 | This study | N/A | |
| Recombinant DNA reagent | pMLINK-PSEN1(F386S)-Aph1-NCT-Pen-2 | This study | N/A | |
| Recombinant DNA reagent | pMLINK-PSEN1(A431E)-Aph1-NCT-Pen-2 | This study | N/A | |
| Recombinant DNA reagent | pMLINK-PSEN1(A434T)-Aph1-NCT-Pen-2 | This study | N/A | |

*Continued on next page*

*Continued*

| Reagent type (species) or resource | Designation | Source or reference | Identifiers | Additional information |
|---|---|---|---|---|
| Recombinant DNA reagent | pET22b-C100-FLAG | In-house | *Lu et al., 2014* | |
| Software, algorithms | Prism 9 version 9.5.1 | GraphPad | https://www.graphpad.com | |
| Software, algorithms | Fiji ImageJ 1.53c | NIH | https://imagej.nih.gov/ | |
| Software, algorithms | AzureSpot | Azure Biosystems | https://azurebiosystems.com/ | |
| Software, algorithms | MassLynx | Waters | https://www.waters.com | |

## Cell lines

Expi293F cells, a HEK cell line purchased from Thermo Fisher, were utilized to produce γ-secretase complexes protein. The stock received from Invitrogen was grown and aliquoted for direct use in these studies without authentication or testing for mycoplasma contamination. HEK293 cells with PSEN1/2 double knockout by CRISPR (*Liu et al., 2021*), a gift of Dr. Lei Liu (Brigham and Women's Hospital, Harvard Medical School, Boston, MA, USA), were used for transfection and FLIM. The cells were authenticated using STR profiling and monitored for mycoplasma contamination every 2 months using MycoAlert Mycoplasma Detection Kit (LONZA, Basel, Switzerland).

## Cell culture conditions

Expi293F cells were cultured in Expi293 expression medium (Thermo Fisher Scientific, A1435101). Transient transfection occurred once the cell density reached $3 \times 10^6$ cells/mL. The cells were kept at 37°C, shaken at 125 rpm, and under 8% $CO_2$. Harvesting took place when cell viability decreased to 75%.

## Bacterial strain and culture

*E. coli* DH5α was employed for molecular cloning and plasmid preparation and was cultured in LB medium at 37°C with constant shaking. *E. coli* BL21 (DE3) was used for the transduction and expression of the γ-secretase substrate C100-Flag (*Li et al., 2000*), grown in LB medium at 37°C with continuous shaking until $OD_{600}$ reached 0.8, at which point expression was induced using 0.5 mM IPTG.

## Methods

### WT and FAD-mutant $\gamma$-secretase constructs

Direct attempts to mutate the PSEN-1 within the vector, according to *Sun et al., 2017*, were unsuccessful due to the large size of the pMLINK tetra-cistronic vector used in this experiment. This plasmid contains genetic codes for all four components of γ-secretase: nicastrin (NCT), presenilin enhancer 2 (Pen2), anterior pharynx-defective 1 (Aph1), and presenilin-1 (PSEN1). To overcome this challenge, we developed a step-by-step LIC method in *E. coli*, combined with restriction digestion of both the insert and vector, which allowed for the successful insertion of mutations. Four monocistronic pMLINK vectors were generated following established methods: pMLINK-PSEN1, pMLINK-Aph1 (bearing a C-terminal HA epitope tag), pMLINK-NCT (with C-terminal V5 and 6XHIS epitope tags), and pMLINK-Pen-2 (featuring N-terminal STREP and FLAG epitope tags) (*Lu et al., 2014*). Each vector contains LINK1 and LINK2 sequences flanking the gene of interest. LINK1 includes a PacI restriction site, while LINK2 contains both PacI and SwaI restriction sites. To create a tricistronic plasmid containing genetic codes for NCT, Pen2, and Aph1, we employed a two-step process. First, we used LIC in *E. coli* to combine NCT and Pen2 through restriction digestion, forming a bicistronic plasmid. The pMLINK-Pen-2 and pMLINK-NCT vectors were treated with restriction enzymes PacI and SwaI, respectively. Both fragments were then electrophoresed through a 1% agarose gel to separate and purify the Pen2 DNA and to linearize and isolate the NCT vector. The Pen2 fragment and the linearized pMLINK-NCT vector were treated with T4 polymerase for 20 min at ambient temperature in the presence of dCTP or dGTP, respectively. The purified T4 polymerase-treated Pen2 fragment was inserted into the purified linearized pMLINK-NCT by LIC to create a bicistronic pMLINK-NCT-Pen-2 vector. Subsequently, we further modified the bicistronic plasmid by including Aph1 through another round of restriction digestion and LIC in *E. coli*, resulting in a tricistronic plasmid. Finally, to create a mutated tetra-cistronic plasmid, we performed multi-site-directed mutagenesis on the PSEN1 and inserted this monocistronic

construct into the tricistronic plasmid through additional rounds of restriction digestion and LIC in *E. coli* (see *Figure 1—figure supplement 1* for illustration).

## γ-Secretase expression and purification

γ-Secretase was produced and purified from Expi293F cells as described in earlier studies (*Bolduc et al., 2017*; *Fraering et al., 2004*; *Osenkowski et al., 2009*). Briefly, Expi293F cells were grown in Expi293 expression medium supplemented with penicillin-streptomycin until they reached a density of $3 \times 10^6$ cells/mL. Before transfection, the medium was exchanged for fresh Expi293 expression medium (without penicillin-streptomycin). For the transfection process, 100 µg of the pMLINK tetracistronic vector and 320 µL of ExpiFectamine 293 Reagent were mixed in 12 mL of Opti-MEM I Reduced Serum Medium and incubated at room temperature for 20 min. The resulting ExpiFectamine 293/plasmid DNA complexes were then added to the cell culture. After 20 hr, ExpiFectamine 293 Transfection Enhancer 1 and Enhancer 2 were added, and the cells were cultured until their viability dropped to 75%. The cells were then harvested by centrifugation at $300 \times g$ for 5 min and resuspended in a buffer containing 50 mM MES (pH 6.0), 150 mM NaCl, 5 mM $CaCl_2$, and 5 mM $MgCl_2$. They were lysed by passing twice through a French press. Unbroken cells and debris were removed by centrifugation at $3000 \times g$ for 10 min, and the supernatant was ultracentrifuged at $100,000 \times g$ for 1 hr to obtain the membrane pellet. The membrane pellet was washed with 0.1 M sodium bicarbonate (pH 11.3) by sequentially passing through syringes with 18-, 22-, 25-, and 27-gauge needles, followed by another ultracentrifugation at $100,000 \times g$ for 1 hr. The pellet was resuspended in 50 mM HEPES (pH 7), 150 mM NaCl, and 1% CHAPSO, and incubated at 4°C with shaking for 1 hr. This mixture was ultra-centrifuged again at $100,000 \times g$. The supernatant was incubated with anti-FLAG M2-agarose beads in TBS containing 0.1% digitonin for 16 hr at 4°C with shaking. The beads were washed three times with TBS/0.1% digitonin before eluting the γ-secretase complex using a buffer with 0.2 mg/mL FLAG peptide in TBS/0.1% digitonin. The eluate was stored at –80°C until needed.

## C100-FLAG expression and purification

Two versions of C100-Flag, light- and heavy-isotope-labeled C100-Flag were prepared. *E. coli* BL21 cells transformed with C100-FLAG construct in pET22b vector (*Li et al., 2000*) were grown in media at 37°C with 180 rpm shaking until $OD_{600}$ reached 0.8. The standard media and M9 minimal media with 20% $^{13}C$-glucose (Cambridge Isotope Laboratories) and $^{15}NH_4Cl$ (Cambridge Isotope Laboratories) were used to produce light-isotope labeling and heavy-isotope-labeled C100-Flag substrate respectively. M9 Minimal media composition: 3.4 g of anhydrous $N_2HPO_4$, 8.794 g of $KH_2PO_4$, 0.25 g of NaCl, 0.5 g of $^{15}NH_4Cl$, 10 mL of 20% $^{13}C$ glucose, 1 mL of 1 M $MgSO_4$-$7H_2O$, 10 µL of 1 M $CaCl_2$-$2H_2O$, 500 µL of 0.5% thiamine-HCl, 5 mL of BME vitamin solution (Sigma-Aldrich), and 10 µL of 1 M $FeSO_4$-$7H_2O$ were dissolved into 500 mL sterilized and deionized water. Cells were induced with 0.5 mM IPTG and cultured for 3 hr. After harvesting by centrifugation, the cells were resuspended in lysis buffer (25 mM Tris, pH 8, and 1% Triton X-100) and lysed by passing through a French press three times. The cleared lysate was incubated with anti-FLAG M2-agarose beads (Sigma-Aldrich) at 4°C for 16 hr with shaking. The beads were washed three times with lysis buffer, and the C100-FLAG protein was eluted using a buffer containing 100 mM glycine at pH 2.7 and 0.25% NP-40, followed by neutralization with Tris buffer at pH 8 and stored at –80°C. The composition and integrity of C100-FLAG were confirmed by SDS-PAGE with western blotting and MALDI-TOF MS.

## Antibodies and western blot analysis for γ-secretase expressed from Expi293F cells and C100 expressed from *E. coli* cells

The following antibodies were used: Anti-Nicastrin (Novus Biologicals, NBP2-57365), Anti-PSEN-1-NTF (Bio-Legend, 823401), anti-PSEN-1-CTF (Cell Signaling, 5643), Anti-Aph-1 (Bio-Legend, 823101), and Anti-Flag M2 (Sigma-Aldrich, F1804). Western blot analysis was carried out using standard procedures. Protein levels of the purified γ-secretase enzyme and the C100 substrate were quantified using a BCA assay (Thermo Fisher Scientific, USA) and normalized for different expression levels based on band intensity. These proteins were separated by SDS-PAGE and transferred onto PVDF membranes. Immunoblotting was then performed and visualized using the enhanced chemiluminescence method.

## In vitro γ-secretase assay

The in vitro γ-secretase assay was conducted as previously detailed (*Devkota et al., 2021*). In summary, 30 nM of either WT or FAD-mutant γ-secretase was preincubated for 30 min at 37°C in an assay buffer

containing 750 µL of 50 mM HEPES (pH 7.0), 150 mM NaCl, 125 µL of DOPC (1,2-dioleoyl-*sn*-glyc ero-3-phosphocholine), and 125 µL of DOPE (1,2-dioleoyl-*sn*-glycero-3-phosphoethanolamine). The reactions were initiated by adding the light- or heavy-isotope form of the C100-FLAG substrate to a final concentration of 5 mM and incubated at 37°C for 16 hr. The reactions were terminated by flash freezing in liquid nitrogen and stored at –20°C.

## Tri- and tetrapeptide quantification by LC-MS/MS

As previously mentioned, small peptides were examined by LC-MS/MS utilizing an ESI quadrupole time-of-flight (Q-TOF) mass spectrometer (Q-TOF Premier, Waters) (*Devkota et al., 2021*). Before being injected into a C18 analytical chromatography column, the reaction mixtures containing light C100-FLAG/WT γ-secretase and heavy C100-Flag/FAD γ-secretase were mixed 1:1. The reaction mixtures were then eluted using a step gradient of 0.08% aqueous formic acid, acetonitrile, isopropanol, and a 1:1 acetone/dioxane mixture. Tandem MS was used to determine which three collision-induced dissociation fragments were the most prevalent for each small peptide. A C18 analytical chromatography column was loaded with different amounts of the synthetic peptide standards (>98% purity, New England Peptide) after they had been dissolved in an assay buffer for LC-MS/MS analysis. Employing an ion mass width of 0.02 unit, the signals from the three most abundant ions were added to produce a peptide chromatographic area. The 'V' mode was used for data acquisition.

## Quantification of AICD species by MS

AICD-FLAG in the reaction mixture was immunoprecipitated with anti-FLAG M2 beads (Sigma-Aldrich) in 10 mM MES pH 6.5, 10 mM NaCl, 0.05% DDM detergent for 16 hr at 4°C. AICD products were eluted from the anti-FLAG beads with acetonitrile: water (1:1) with 0.1% trifluoroacetic acid. In parallel, different concentrations of AICD standards (2500 nM, 2000 nM, 1000 nM, 500 nM, 250 nM, 125 nM, and 62.5 nM) were prepared using synthetic peptides. To all samples and standards, 5 nM of ProteoMass Insulin MALDI-MS was added as an internal standard. Finally, the elutes were analyzed on a Bruker Autoflex MALDI-TOF MS.

## Quantification of Aβ40 and Aβ42 by ELISA

To quantify Aβ peptides, we analyzed reactions from in vitro cleavage assays involving purified γ-secretase complexes and the C100-FLAG substrate using specific ELISA kits from Invitrogen, in accordance with the manufacturer's instructions.

## Fluorescence lifetime imaging microscopy

Fluorescence emissions were collected using the ET525/50m-2p filter (Chroma Technology Corp, Bellows Falls, VT, USA). The donor Alexa Fluor 488 lifetime was recorded using a high-speed photomultiplier tube (MCP R3809; Hamamatsu photonics, Hamamatsu City, Japan) and a time-correlated single-photon counting acquisition board (SPC-830; Becker & Hickl GmbH, Berlin, Germany). The acquired FLIM data were analyzed using SPC Image software (Becker & Hickl GmbH). Pseudo-colored images corresponding to the ratios of 6E10-Alexa Fluor 488 over C99-720 emission were generated in MATLAB (MathWorks, Natick, MA, USA).

## Materials availability statement

Materials are available from the authors upon request.

# Acknowledgements

We thank L Liu (Harvard Medical School/Brigham and Women's Hospital) for HEK293 cells with PSEN1/2 doubly knocked out through genome editing. This work was supported by US. National Institutes of Health (NIH) grants AG66986 (to MSW) and AG079569 (to J Chhatwal; Co-PI MSW). P Arafi was supported by an undergraduate research award from NIH grant P20 GM103418.

## Additional information

### Funding

| Funder | Grant reference number | Author |
| --- | --- | --- |
| National Institute on Aging | AG66986 | Michael S Wolfe |
| National Institute on Aging | AG79569 | Michael S Wolfe |

The funders had no role in study design, data collection and interpretation, or the decision to submit the work for publication.

### Author contributions

Parnian Arafi, Data curation, Formal analysis, Investigation, Methodology; Sujan Devkota, Formal analysis, Supervision, Investigation, Methodology; Emily Williams, Formal analysis, Investigation; Masato Maesako, Conceptualization, Data curation, Formal analysis, Supervision, Investigation, Methodology; Michael S Wolfe, Conceptualization, Formal analysis, Supervision, Investigation, Project administration

### Author ORCIDs

Masato Maesako ⓘ https://orcid.org/0000-0002-1970-2462
Michael S Wolfe ⓘ https://orcid.org/0000-0002-5721-9092

Reviewer #1 (Public review): https://doi.org/10.7554/eLife.102274.3.sa1
Reviewer #2 (Public review): https://doi.org/10.7554/eLife.102274.3.sa2
Author response https://doi.org/10.7554/eLife.102274.3.sa3

## Additional files

### Supplementary files

Supplementary file 1. Table of cross-reactivity of Aβ43 peptide with Aβ42 ELISA kit. Cross-reactivity of Aβ43 with Aβ42 in ELISAs. Various concentrations of Aβ43 (ranging from 15.63 pg/mL to 1,000,000 pg/mL) were tested using Aβ42-specific ELISA kits. The instrument readings for each concentration are displayed, indicating significant cross-reactivity starting at 250 pg/mL (0.06 nM) of Aβ43. (Note: 'OF' stands for overflow.)

Supplementary file 2. Table of cross-reactivity of Aβ43 peptide with Aβ40 ELISA kit. Cross-reactivity of Aβ43 with Aβ40 in ELISAs. Different concentrations of Aβ43 (ranging from 7.8 to 1,000,000 pg/mL) were assessed using ELISA kits specific for Aβ40. The resulting instrument readings for each concentration are presented, revealing cross-reactivity beginning at 500 pg/mL (0.12 nM) of Aβ43.

MDAR checklist

### Data availability

*Figure 1—source data 1*, *Figure 2—source data 1*, *Figure 3—source data 1*, *Figure 4—source data 1*, and *Figure 4—figure supplement 1—source data 1* contain the numerical data used to generate the figures.

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
