## [Editor Report · eLife Assessment]

This manuscript provides **fundamental** studies to gain insight into the mutations in the presenilin-1 (PSEN1) gene on proteolytic processing of the amyloid precursor protein (APP). The authors provide **compelling** evidence using mutations in PSEN to understand what drives alternative substrate turnover with **convincing** data and rigorous analysis. This deep mechanistic study provides a framework towards the development of small molecule inhibitors to treat AD.

---

## [Referee Report · Reviewer #1 (Public review)]

Summary:

Arafi et al. present results of studies designed to better understand the effects of mutations in the presenilin-1 (PSEN1) gene on proteolytic processing of the amyloid precursor protein (APP). This is important because APP processing can result in the production of the amyloid β-protein (Aβ), a key pathologic protein in Alzheimer's disease (AD). Aβ exists in various forms that differ in amino acid sequence and assembly state. The predominant forms of Aβ are Aβ40 and Aβ42, which are 40 and 42 amino acids in length, respectively. Shorter and longer forms derive from processive proteolysis of the Aβ region of APP by the heterotetramer β-secretase, within which presenilin 1 possesses the active site of the enzyme. Each form may become toxic if it assembles into non-natively folded, oligomeric, or fibrillar structures. A deep mechanistic understanding of enzyme-substrate interactions is a first step toward the design and successful use of small-molecule therapeutics for AD.

The key finding of Arafi et al. is that PSEN1 amino acid sequence is a major determinant of enzyme turnover number and the diversity of products. For the biochemist, this may not be surprising, but in the context of understanding and treating AD, it is immense because it shifts the paradigm from targeting the results of γ-secretase action, viz., Aβ oligomers and fibrils, to targeting initial Aβ production at the molecular level. It is the equivalent of taking cancer treatment from simple removal of tumorous tissue to the prevention of tumor formation and growth. Arafi et al. have provided us with a blueprint for the design of small-molecule inhibitors of γ-secretase. The significance of this achievement cannot be overstated.

Strengths and weaknesses:

The comprehensiveness and rigor of the study are notable. Rarely have I reviewed a manuscript reporting results of so many orthogonal experiments, all of which support the authors' hypotheses, and of so many excellent controls. In addition, as found in clinical trial reports, the limitations of the study were discussed explicitly. None of these significantly affected the conclusions of the study.

Some minor concerns were expressed during the review process. The authors have revised the manuscript, and in doing so, dealt appropriately with the concerns and strengthened the manuscript.

---

## [Referee Report · Reviewer #2 (Public review)]

Summary:

The work by Arafi et al. show the effect of Familial Alzheimer's Disease presenilin-1 mutants on endoproteinase and carboxylase activity. They have elegantly demonstrated how some of mutants alter each step of processing. Together with FLIM experiments, this study provides additional evidence to support their 'stalled complex hypotheses'.

Strengths:

This is a beautiful biochemical work. The approach is comprehensive.

Weaknesses:

However, the novelty of this manuscript is questionable since this group has published similar work with different mutants (Ref 11) .

---

## [Author Response]

The following is the authors’ response to the original reviews.

comprehensiveness and rigor of the study are notable. Rarely have I reviewed a manuscript reporting the results of so many orthogonal experiments, all of which support the authors' hypotheses, and of so many excellent controls.” Reviewer 2 commented: “They have elegantly demonstrated how some mutants alter each step of processing. Together with FLIM experiments, this study provides additional evidence to support their 'stalled complex hypotheses'….This is a beautiful biochemical work. The approach is comprehensive.”

Below we respond to the relatively minor concerns of Reviewer 2, which may be included with the first version of the Reviewed Preprint.

**Reviewer 2:**
(1) It appears that the purified γ-secretase complex generates the same amount of Aβ40 and Aβ42, which is quite different in cellular and biochemical studies. Is there any explanation for this?

Roughly equal production of Aβ40 and Aβ42 is a phenomenon seen with purified enzyme assays, and the reason for this has not been identified. However, we suggest that what is meaningful in our studies is the relative difference between the effects of FAD-mutant vs. WT PSEN1 on each proteolytic processing step. All FAD mutations are deficient in multiple cleavage steps in γsecretase processing of APP substrate, and these deficiencies correlate with stabilization of E-S complexes.

(2) It has been reported the Aβ production lines from Aβ49 and Aβ48 can be crossed with various combinations (PMID: 23291095 and PMID: 38843321). How does the production line crossing impact the interpretation of this work?

In the cited reports, such crossover was observed when using synthetic Aβ intermediates as substrate. In PMID 2391095 (Okochi M et al, Cell Rep, 2013), Aβ43 is primarily converted to Aβ40, but also to some extent to Aβ38. In PMID: 38843321 (Guo X et al, Science, 2024), Aβ48 is ultimately converted to Aβ42, but also to a minor degree to Aβ40. We have likewise reported such product line “crossover” with synthetic Aβ intermediates (PMID: 25239621; Fernandez MA et al, JBC, 2014). However, when using APP C99-based substrate, we did not detect any noncanonical tri- and tetrapeptide co-products of Aβ trimming events in the LC-MS/MS analyses (PMID: 33450230; Devkota S et al, JBC, 2021). In the original report on identification of the small peptide coproducts for C99 processing by γ-secretase using LC-MS/MS (PMID: 19828817; Takami M et al, J Neurosci, 2009), only very low levels of noncanonical peptides were observed. In the present study, we did not search for such noncanonical trimming coproducts, so we cannot rule out some degree of product line crossover.

(3) In Figure 5, did the authors look at the protein levels of PS1 mutations and C99-720, as well as secreted Aβ species? Do the different amounts of PS1 full-length and PS1-NTF/CTF influence FILM results?

FLIM results depend on the degree that C99 and long Aβ intermediates are bound to γ-secretase compared to unbound C99 and Aβ. The 6E10-Alexa 488 lifetime is significantly decreased by FAD mutations compared to WT PSEN1 (Fig. 5). However, the observed decrease in lifetime with the PSEN1 FAD mutants might also be due to lower levels of C99-720 expression or higher levels of PSEN1 CTF (i.e., mature γ-secretase complexes). We checked the C99-720 fluorescence intensities in the FLIM experiments and found that C99-720 intensities are not significantly different between cells transfected with WT and those with FAD PSEN1. Furthermore, Western blot analysis shows that levels of C99-720 are not significantly low and those of PSEN1 CTF are not high in FAD PSEN1 compared to WT PSEN1 expressing cells. Although PSEN1 CTF levels trend low for PSEN1 F386S, this mutant resulted in decreased FLIM only in Aβ-rich regions. Thus, the reduced FLIM apparently reflects effects of FAD mutation on E-S complex stability. Levels of full-length PSEN1 were also determined and found not to correlated with FLIM effects, although full-length PSEN1 represents protein not incorporated into full active γ-secretase complexes and therefore does not interact with C99-720.

(4) It is interesting that both Aβ40 and Aβ42 Elisa kits detect Aβ43. Have the authors tested other kits in the market? It might change the interpretation of some published work.

We have not tested other ELISA kits. Considering our findings, it would be a good idea for other investigators to test whatever ELISAs they use for specificity vis-à-vis Aβ43.